# Wheat Omics: Advancements and Opportunities

**DOI:** 10.3390/plants12030426

**Published:** 2023-01-17

**Authors:** Deepmala Sehgal, Priyanka Dhakate, Heena Ambreen, Khasim Hussain Baji Shaik, Nagenahalli Dharmegowda Rathan, Nayanahalli Munireddy Anusha, Rupesh Deshmukh, Prashant Vikram

**Affiliations:** 1International Maize and Wheat Improvement Center (CIMMYT), El Batán, Texcoco 56237, Mexico; 2Syngenta, Jealott’s Hill International Research Centre, Bracknell, Berkshire RG42 6EY, UK; 3National Institute of Plant Genome Research, Aruna Asaf Ali Marg, New Delhi 110076, India; 4School of Life Sciences, University of Sussex, Brighton BN1 9RH, UK; 5Faculty of Agriculture Sciences, Georg-August-Universität, Wilhelmsplatz 1, 37073 Göttingen, Germany; 6Indian Agricultural Research Institute (ICAR-IARI), New Delhi 110012, India; 7Corteva Agriscience, Hyderabad 502336, Telangana, India; 8Department of Biotechnology, Central University of Haryana, Mahendragarh 123031, Haryana, India; 9Bioseed Research India Ltd., Hyderabad 5023324, Telangana, India

**Keywords:** omics, wheat, genomics, transcriptomics, proteomics, metabolomics, multiomics

## Abstract

Plant omics, which includes genomics, transcriptomics, metabolomics and proteomics, has played a remarkable role in the discovery of new genes and biomolecules that can be deployed for crop improvement. In wheat, great insights have been gleaned from the utilization of diverse omics approaches for both qualitative and quantitative traits. Especially, a combination of omics approaches has led to significant advances in gene discovery and pathway investigations and in deciphering the essential components of stress responses and yields. Recently, a Wheat Omics database has been developed for wheat which could be used by scientists for further accelerating functional genomics studies. In this review, we have discussed various omics technologies and platforms that have been used in wheat to enhance the understanding of the stress biology of the crop and the molecular mechanisms underlying stress tolerance.

## 1. Introduction

Wheat is a widely grown staple cereal crop covering an area of 217 million ha annually and meets the dietary demands of 2.5 billion people globally [1]. Its production is severely affected by biotic and abiotic stresses throughout its production zones [2,3,4] leading to a large gap between the potential yield and the real harvest yield. The projected global demand for wheat by 2050 [5] demands that scientists and breeders should adopt newer tools and technologies to accelerate the development of high-yielding varieties that can adapt to challenging environments.

With the recent advancements in wheat omics, which include genomics, transcriptomics, metabolomics and proteomics, extensive research has been conducted to decipher the mechanism(s) of stress tolerance, leading to the enhanced understanding of the expression of different genes along with their protein profiling and the biological mechanisms underlying various stress tolerance traits [6,7]. To handle massive data points generated by all omics approaches, new analytical tools, high-throughput data analysis pipelines and omics databases have been developed [8,9,10,11]. For example, a Wheat Omics database has been recently developed, which combines multiple omics data gathered from different tissues and germplasm sets harboring more than 100,000 novel annotated transcripts [10]. It has various user-friendly tools, including BLAST, Primer Server, Homolog Finder, IntervalTool, etc., and various functionalities to investigate gene expression, co-expression, protein interactions and synteny. WheatGmap database (https://www.wheatgmap.org, accessed on 10 October 2022) was recently developed, which integrates multiple mapping models and large amounts of public data to facilitate functional genomics research [11]. It contains more than 3500 next-generation sequencing (NGS) datasets, including whole genome sequencing (WGS), whole exome sequencing (WES) and transcriptome deep-sequencing (RNA-seq) datasets. These recent developments will allow wheat scientists to further enrich the knowledge of differentially regulated genes under various stress conditions and unravel the complex regulatory interplay of different biological processes in response to multiple stresses in fields. It is anticipated that these advances will play pivotal roles in wheat breeding and trait optimization.

In the past few years, the use of multi-omics has become a leading approach in elucidating the responses to biotic and abiotic stresses and in building prediction models in crops, thus allowing multidimensional research in several crops, including wheat [12,13]. Figure 1 illustrates different omics approaches that are generally deployed in crops. This review highlights various omics/multi-omics approaches and technological advancements that have been utilized in wheat for identifying genes underpinning stress tolerance, investigating the biochemical changes in response to stress, understanding protein–protein interactions and regulating metabolic pathways evolved to tolerate and/or escape stresses. Various case studies have been discussed at length under different sections to better understand the methodology in detail and to realize how the research outputs from omics/multi-omics research could be used in breeding.

## 2. Genomic Approaches

Advancement in NGS technologies has propelled a massive wave of scientific discoveries, providing enormous modern genomic tools and resources for wheat genetic improvement and bringing a paradigm shift in breeding methods [14]. Innovations have been made in the development of efficient genotyping platforms in wheat which could be combined with physical maps for rapid gene discovery [15,16,17]. Advancements in single nucleotide polymorphism (SNP) genotyping arrays now allow thousands of markers to be screened in large populations, providing tangible information in an expeditious and inexpensive mode. Over the years, several SNP arrays have been developed for genotyping wheat, showing continuous improvement in terms of their effectiveness to differentiate diverse wheat genotypes. A 9K iSelect array was developed based on gene-associated SNPs and assessed the functional diversity of 2994 wheat accessions [18]. Additionally, a high-density 90K SNP chip was developed that was subsequently used for the generation of a 15K Infinium array [19,20]. An Affymetrix Axiom 820K SNP array, targeted the identification of polymorphisms in bread wheat and close relatives, including members from secondary and tertiary gene pools [21]. This array was further exploited to derive a 35K Wheat Breeders Array which was found to be highly equipped to genotype elite wheat accessions, including landraces from breeding programs [22]. Whole genome resequencing data from eight wheat lines were employed to develop a 280K SNP array for effective discrimination between closely related wheat accessions [23].

Recently, previously reported seven wheat SNP arrays were evaluated to design a cumulative 660K Axiom SNP array with wider applicability [17]. Based on a novel approach, an imputation-enabled SNP array, Illumina Infinium Wheat Barley 40K SNP array version 1.0, has been designed to capture the haplotypic diversity in wheat and barley germplasm [24]. In addition to the reliable and cost-effective SNP chip arrays, NGS-based genotyping-by-sequencing (GBS) has been used to detect global variation in wheat germplasm for various applications [25,26,27,28,29,30,31,32,33,34].

In the past decade, several reference genomes sequences have been released for wheat, including durum wheat (*Triticum turgidum* ssp. *durum*;) [35,36] bread wheat (*T. aestivum*) [9,37,38,39,40,41,42,43,44] and its progenitor species including wild emmer (*T. turgidum* spp. *dicoccoides*) [35] wild goatgrass (*Aegilops tauschii*) [45,46,47] and *T. urartu* [48]. These near-complete assemblies, for the first time, depicted the true genomic landscape of wheat and provided genome-wide navigation into repeat and gene families, transforming the entire field of wheat genomics. The availability of multiple high-quality reference genomes and resequencing data from several genotypes in wheat has revealed a high proportion of genomic variation [40]. Moreover, an awareness that a single reference genome is inadequate to represent the complete diversity within the species has catalyzed the expansion of a pan-genomic era in wheat. Pan-genomes aim at investigating the entire sequence diversity within a species, captured from the core/universal (present in all individuals) and divergent/variable (specific to a few individuals) genomic components, which could potentially be applied for understanding the genetic basis of important agronomic traits [49]. The first wheat pan-genome, based on the gene content of 18 cultivars, identified a new gene space along with the exploration of variable genes through presence/absence variation (PAV) analysis with respect to the classical reference genome of Chinese Spring [50]. Functional annotation of the variable gene components of the studied wheat cultivars exhibited the enrichment of genes involved in the response to abiotic and biotic stresses. The dispensable gene sets were shown to harbour more genomic variation as compared to core genes and were projected to play an important role in maintaining crop diversity.

The reference genome assemblies have served as a key component for in-silico identification and characterization of known and novel gene families with functional implications in various biotic and abiotic stresses. For instance, examination of nucleotide-binding leucine-rich repeats (NLR) gene family revealed a highly variable and complex repertoire of the NLR-encoding resistance gene family [40]. Only 30–34% of NLR signatures were conserved among the studied lines and a large number of unique signatures were reserved in individual genotypes, providing high-resolution variation information which could be exploited for the development of disease-resilient wheat cultivars. Additionally, exploration of a novel gene family facilitates the identification of gene members, analysis of their gene structure and duplication events, evolutionary relationships, spatial and temporal expression dynamics and assignment of putative function. A plethora of studies have been conducted to elucidate the novel functional roles of different gene families during various stress conditions in wheat. Reporting the entire set of these studies would be difficult to accommodate here, owing to the excessively high quantum; however, a few representative studies are presented in Table 1.

The availability of millions of SNPs in wheat has opened vistas of new gene discoveries using genome-wide association studies (GWAS) [14,73]. As of today, more than 3000 marker-trait associations are known for a plethora of agronomic traits in wheat which have been compiled into 141 meta-QTLs [74]. Thirteen of these meta-QTLs have been reported as breeder’s meta-QTLs, the application of which in breeding should prove useful for realizing yield gains. Due to the high marker densities available in wheat, haplotypes-based GWAS studies and epistatic interaction analyses have become possible in wheat, providing high-resolution dissection of complex traits [75,76,77,78,79,80,81,82,83,84]. Additionally, a range of new genetics and genomics analyses are now possible, including genomic selection (GS) and selective sweep analyses, which require genome-wide high-density markers [85,86,87,88]. The integration of genomics in wheat improvement programs has restructured breeding programs globally, from conventional to genomics-assisted ones, promoting marker-assisted selection (MAS) and GS for achieving faster and more reliable breeding outcomes [89].

Due to its intrinsic ability to expedite genetic gains achieved through improved selection accuracy coupled with reduced breeding time and phenotyping cost, GS has garnered huge attention over the past decade in wheat [85]. Various efforts have been undertaken to practically implement the GS strategy for abiotic stress tolerance, particularly heat and drought stress, and different robust statistical models have been tested to analyze their impact on prediction accuracies [77,90,91,92,93,94,95,96]. Additionally, attempts have been made to optimize genomic prediction accuracies for resistance against a wide range of pathogens in wheat, including powdery mildew (causative agent: *Blumeria graminis*; [97]), fusarium head blight (causative agent: *Fusarium graminearum*; [98,99,100,101]), septoria tritici blotch (causative agent: *Zymoseptoriatritici*; [99,102,103]), stem or black rust (causative agent: *Puccinia graminis* Pers; [104]), leaf or brown rust (causative agent: *Puccinia triticina* Eriks; [104]), stripe or yellow rust (causative agent: *Puccinia striiformis* West; [104,105]), stagonospora nodorum blotch (causative agent: *Parastagonospora nodorum*; [102]), spot blotch (causative agent: *Bipolaris sorokiniana*; [32] and tan spot (causative agent: *Pyrenophora tritici-repentis*; [102]). Integrating omics datasets in prediction models and compounding GS with high-throughput phenotyping, speed breeding and gene editing techniques would be instrumental in hastening wheat improvement programs for the development of future climate-resilient, high-yielding superior wheat cultivars.

## 3. Transcriptomic Approaches

In the past, microarrays have been used extensively to analyze the expression and co-expression of numerous genes under various stress conditions in crop plants [106,107,108]. However, microarray-based experiments failed to detect gene networks regulating stress responses at the genome-wide level. With advancements in NGS, whole transcriptome analyses have become feasible, allowing identification and quantification of the global expression of transcripts, alternative splicing patterns and associated allele-specific expressions [109,110]. RNA-seq, the latest NGS technique for investigating genome-wide transcriptomes, helps to examine the expressional variation in genes in contrasting sets of samples or panels subjected to different stress treatments and to pinpoint the potential candidate genes. Due to its in-depth coverage and global expression of transcripts, RNA-seq has been used extensively in many crops, including wheat, to uncover the mechanisms conferring tolerance to different stresses. Table 2 enlists transcriptomic studies carried out in wheat under different stress conditions in the past decade.

Transcriptomics studies have unveiled important roles played by numerous genes, gene families, transcription factors (TFs), hormones, metabolites, cofactors and microRNAs (miRNAs) in conferring tolerance to various abiotic stresses in wheat. Most significantly, the role of antioxidant enzymes, such as cytochrome P450, glutathione S-transferase, polyphenol oxidase, chitinase 2, ascorbate peroxidase (APX) and peroxidase (POD), in circumventing the adverse effects of reactive oxygen species (ROS) on proteins has been repeatedly suggested to be an important tolerance mechanism under both drought and salinity stress conditions [122,123,124]. Similarly, ABC transporters and Na^+^/Ca^2+^ exchangers have been suggested to be the key players under salinity stress conditions. In addition, the role of various TFs belonging to the NAC, WRKY and MADS families was highlighted to be a significant one under a single or multiple abiotic stress condition while TFs from the MYB family were highlighted as the key candidate genes under biotic stress conditions [125,126].

The transcriptomic alterations in response to salt stress were analysed to identify the candidate genes regulating plant salt stress tolerance in wheat [125]. Two sets of genotypes, contrasting for sensitivity to salinity stress wherein one set consisted of a German winter wheat cultivar Zentos (salt-tolerant) and the synthetic genotype Syn86 (salt-susceptible), while the other set had a Turkish cultivar Altay2000 (salt-tolerant) and the Uzbek cultivar Bobur (salt-susceptible) were analysed [125]. The expression of genes from four genotypes revealed upregulation of some ABC transporters and Na^+^/Ca^2+^ exchangers in the tolerant genotypes, indicating their involvement in the mechanisms of sodium exclusion and homeostasis. Furthermore, five genes were found co-located with two QTLs on chromosome 2A and three of these were found to have a differential expression. Of these, TraesCS2A02G395000, which codes for an oxoglutarate/iron-dependent dioxygenase, was reported to be central in controlling salinity stress in wheat [125]. Transcriptomic analyses in wheat was employed to assess regulation of biosynthesis of a special metabolite, benzoxazinoid, by the wheat transcription factor *MYB31* in response to pest attack [126]. The silencing of *TaMYB31* gene was found to significantly decrease the benzoxazinoid metabolite levels and consequently resulting in susceptibility to herbivores [126]. Furthermore, comprehensive transcriptomics highlighted the fact that the *TaMYB31* gene co-expresses with the target benzoxazinoid-encoded *Bx* genes under various biotic and environmental circumstances imparting resistance to wheat plants [126]. RNAseq analysis of wheat lines subjected to multiple abiotic stresses revealed a novel ERF gene TaERF-6-3A, which was found to be induced under three abiotic stresses, salt, cold and drought [127]. In addition, upregulation of 20 *AP2/ERF* genes was identified based in response to drought stress [127]. Most significantly, a novel *ERF* gene, *TaERF-6-3A*, was found to be induced in response to three abiotic stresses, salt, cold and drought [127]. Further, the ectopic expression of *TaERF-6-3A* in *Arabidopsis* was found to increase sensitivity to drought and salinity stress. *TaERF-6-3A* was found to repress endogenous antioxidant enzyme mechanisms, resulting in increased oxidative damage and reduced tolerance to stress, suggesting *TaERF-6-3A* to be a negative regulator [127].

Over the last decade, transcriptomics has validated the regulatory role of miRNAs in response to abiotic stresses in wheat [128,129,130,131]. The microRNAs (miRNAs) are small, single-stranded, non-coding RNAs with a typical length of 20–24 nucleotides. The role of miRNAs as cardinal post-transcriptional regulators directing gene expression in a sequence-specific manner in response to abiotic stresses is now well established. Stress-induced expression of miRNA genes usually leads to the accumulation of mature miRNA species to downregulate their intended target. Alternatively, the downregulation of these genes results in an increased expression in their intended targets. Over the past decade, transcriptomics has been used to assess the altered expressions of miRNAs and their downstream target in response to numerous abiotic stresses. RNA-seq was employed in the salt-tolerant wheat cultivar (Arg) and salt-susceptible cultivar (Moghan3) and identified a total of 26,171 novel transcripts, of which 5128 were found to be differentially expressed in response to salinity stress [129]. Significant expression of 16 novel miRNAs was found in response to salinity stress, in addition to the *TaABAC15*, *TaACA7*, *TaANN4* and *TaNCL2* genes, in the salt-tolerant wheat cultivar, implicating a central role of miRNA in mediating responses to salinity stress in wheat. A root-specific accumulation of mature miRNAs under salinity stress in wheat was identified [131]. Most importantly, miRNAs *osamiR444b.2*, *osamiR172b* and *athmiR5655* were found to mediate the regulation of TFs, containing conserved domains such as MADS-box, AP2/ERBP and bHLH135, respectively [131].

Furthermore, underlying regulatory network at play to improve tolerance to heat was explored with the exogenous application of trehalose in wheat [124]. It was found that pretreatment with trehalose was found to alleviate the oxidative damage caused by high-temperature stress [124]. A total of 15,403 genes were found differentially expressed in the tolerant Yangmai 18 variety and the differential expression of 749 genes was found in the susceptible Yannong 19, suggesting that a higher number of genes are involved in the trehalose response in the tolerant variety [124]. A majority of the DEGs were found to mediate cellular metabolism and regulation cell metabolism and biological regulation. Additionally, RNA-seq analysis revealed that the exogenous application of trehalose could potentially activate autophagy under high-temperature stress. Conclusively, the application of trehalose in wheat plants was found to reset the transcriptional profile of the endogenous metabolic pathways, suggesting the existence of a trehalose-induced complex regulatory network at play to aid the efficient adaptation of plants in response to heat stress [124].

Numerous transcriptomic studies have suggested a co-mediated regulation of Calmodulin (CaM) and calmodulin-like (CML) proteins regulating Ca^2+^-signaling processes in response to abiotic stresses in wheat [51,132]. The transcriptome from leaves of the salt-tolerant Iranian wheat cultivar (Arg) was analysed on Illumina HiSeq 2500 platform [129]. A total of 26,700 novel transcripts along with the differential expression of 4290 genes, were identified, of which 2346 were found upregulated and 1944 were downregulated [129]. Transcripts discovered were primarily found to be related to phenylpropanoid biosynthesis, transporters, exosomes, transcription factors, MAPK signalling, glycosyltransferases and signal transduction, suggesting their relevance in imparting salt tolerance to the plant [129]. Significant upregulation of Ca^2+^ homeostasis-related genes, including two genes for calcium-transporting ATPases, three glutamate receptors (GLRs), 13 CIPK genes, a gene encoding calmodulin (CaM), a CaM-binding transcription factor *CBP60* and a CaM-binding gene *TaMLO* was found [129]. In addition, six wall-associated kinases (WAKs) and three LecRLKs, which are both members of the receptor-like kinases (RLK) subfamily, were discovered to be upregulated [129]. Likewise, LecRLKs and WAKs have a well-documented role in controlling plants’ adaptation to various abiotic stresses [133]. In addition, 27 MYBs, basic leucine zipper TFs, 17 zinc finger TFs and two genes coding for TIFY were found to be differentially expressed in response to salinity stress in wheat [129]. Further, 22 genes, reported to be involved in regulating the transportation of sodium, potassium or both, were found to be significantly upregulated in response to salt stress [129]. Of these, two genes, *Ta.HKT1* and *Ta.KT*, were functionally characterized. *Ta.HKT1* was established as an orthologue of rice Os04g0607500 that encodes a cation transporter. *Ta.KT* was found to be an orthologue of At2g30070 that encodes AtKT1, a potassium transporter. Transcripts of both genes were found to be significantly accumulated in response to salinity stress [129]. Similarly, recently 113 *TaCMLs* and 15 *TaCAMs* were identified in response to drought and salinity stress [51]. Increased expression of *TaCAM2-D* was found under salt and drought stresses [51]. Evidently enough, the ectopic expression of *TaCAM2-D* in Arabidopsis led to an increased tolerance to salt and drought. *TaCAM2-D* was validated to be *TaMPK8*, a wheat mitogen-activated protein kinase, providing evidence of calcium signalling in regulating the stress response in plants [51].

Heavy metal contamination of the soil via natural and (or) anthropogenic activities is another stress that crop plants regularly experience. The physical-geochemical properties of these heavy metals in soil and their subsequent uptake by plants have been found to adversely affect various physiological, morphological and biochemical processes of the plant, resulting in reduced crop productivity. Given the relevance of wheat as an important cereal across the globe, developing/screening wheat cultivars with low heavy metal accumulation promises reduced dietary exposure to heavy metals. To this effect, root transcriptomes of YM16, a low-Cadmium (Cd)-accumulating wheat genotype were analysed under cadmium-treated and untreated conditions [134]. Genes regulating glutathione metabolism, phenylpropanoid biosynthesis, nitrogen metabolism and sulfur metabolism were found to be significantly upregulated under Cd stress [134]. Furthermore, a comparative transcriptomic approach was employed to assess the activation of endogenous Cd detoxification mechanisms following Cd treatment in wheat genotypes T207 and S276 displaying Cd resistance and sensitivity, respectively [135]. The accumulation of transcripts encoding glutathione, ascorbic acid and catalase activity was found in T207 along with lower expression levels of peroxidase and superoxide dismutase [135]. Further, the transcriptomics revealed that the expression of Respiratory Burst Oxidase Homologs (RBOH), such as *RBOHA*, *RBOHC*, and *RBOHE*, was significantly increased in response to Cd toxicity [135]. Twenty-two *RBOH* genes were found to have increased expression levels in S276 in comparison with T207. Studies, as mentioned above, continue to increase our understanding of the complex molecular mechanisms of plants under Cd stress and shall aid in the development of low-heavy metal accumulating wheat varieties.

## 4. Metabolomic Approaches

Metabolomics is one of the most evolved and extensively explored omics technologies. It is an analytical profiling platform to measure and compare the metabolites present in biological samples at a given time. Metabolomics can be combined with high-throughput analytical chemistry and multivariate data analysis to unravel the molecular mechanisms at play in response to endogenous and exogenous environments of organisms. Its ability to efficiently complement other omics approaches makes it even more powerful for usage across all organisms [74]. In contrast to other omics approaches, metabolomics is efficient in tangibilizing the biochemical activities of apparent proteins and their subsequent metabolism into other proteins, making it rather easy to associate phenotypes [74]. Albeit not as extensively as transcriptomics, metabolomics has been employed in wheat to identify stress tolerance mechanisms and candidate genes by correlating the accumulation of metabolites in response to induced stresses [41,136,137]. An untargeted metabolomic analysis was employed to validate the use of wheat homeologous group 3 ditelosomic lines for identification and direct validation of genes regulating the accumulation of metabolites during the later stages of grain development [136]. Significant differences with respect to the accumulation of sugars were found between genetically modified wheat genotypes with their parental lines associated with the environment rather than the genotype [138]. Similar findings have also been reported in other investigations [137,139,140]. To assess the effects of nutrient deficiency on the metabolomic profile of wheat, a metabolomic approach in conjunction with transcriptomics was employed to assess the effects of nitrogen and sulphur deficiency on the remobilization of resources from a degrading canopy to developing grains in wheat [141]. The results obtained from both approaches corroborated, to suggest a suboptimal mobilization of N and (or) S supply to leaves but the supply to developing grains was always found to be optimum. A clever strategy employed by the plant was deciphered that utilizes the machinery in place for aminoacid biosynthesis to produce glutamine in developing grains. In the first seven days of seed development, a significant accumulation of glutamine was observed that was later converted to other amino acids and proteins over the subsequent 21 days of grain development [141]. The content of nitrogen and sulphur in the grains was found to increase at a steady rate of post-anthesis and nitrogen deficiency was found to adversely affect the accumulation of nitrogen and sulphur, suggesting that the availability of nitrogen at the vegetative stage determines the time and extent of the remobilization of resources [141].

The metabolic profile of the wheat cultivar Sumai-3 was analysed to unravel the host resistance mechanisms against Fusarium Head Blight (FHB) in response to two isolates of *Fusarium graminearum* that differentiated with regard to the production of trichothecene [142]. The resistance was found to be acquired due to the accumulation of the resistance-related (RR) metabolites from the phenylpropanoid pathway that reduced the spread of fungus by thickening the host cell wall and not due to the reduced accumulation of the toxic deoxynivalenol (DON) [142]. In addition, the growth of the pathogen was found curbed owing to the antifungal properties of RR phenylpropanoids that subsequently decreased the biosynthesis of trichothecene. Evidently, these results demonstrated the efficacy of the metabolomic approach in dissecting the underlying mechanisms driving biotic stress tolerance in wheat [142].

An increased concentration of atmospheric CO_2_ has been reported to have an adverse effect on the metabolomic profile of the wheat plant. A change in 40 metabolites was found with an increase in atmospheric CO_2_ levels that was associated with the altered development of the plant [143]. This change was correlated with the decreased concentration of a few amino acids and derivatives that induce the raffinose synthesis pathway only during the vegetative phase but a reduced lysine turnover throughout the plant’s life cycle. Similarly, increased atmospheric CO_2_ was found to be associated with a decrease in the accumulation of N-rich metabolites vis-a-vis a few organic acids and ribose-5-P [144]. Interestingly, soon after, one of the first reports of integrating the metabolomic approach with other high-throughput studies was reported. The genetic basis of variation in metabolites and agronomic traits was dissected by integrating the mapping of QTLs associated with the levels of metabolites with the mapping of the agronomic traits [145]. Furthermore, a genetic control for numerous groups of metabolites was found to be regulated by two distinct loci on chromosome 7A. Such a study is a great example of the integration of metabolomics with QTL analysis to identify potential trait targets with accuracy.

Metabolomics has also been employed to investigate salinity tolerance in durum wheat [146]. Adaptation to salinity was reported acquired by rearranging sucrose and nitrogen-containing metabolites in cytosol following the uptake of sodium and its sequestration in the vacuoles. Such a strategy ensures the optimal maintenance of osmoticum to prevent any oxidative damage to the root cell and ensures plant survival under nitrogen-deficient conditions. Under low nitrate conditions, accumulation of glycine betaine and sucrose was found [146]. However, under high nitrate conditions, glycine betaine, proline and asparagine were found to maintain cytosolic osmoticum, scavenge ROS and assimilate excess ammonium [146]. Differential changes in the metabolomic profiles, correlated with physiological changes, were found in leaves and roots of two wheat genotypes, LA54 and AGS2038, drought-tolerant and drought-susceptible, respectively, to assess the drought-responsive mechanisms in wheat [147]. In LA54, 45 metabolites in the leaves were found to be altered compared to only 20 in the roots, suggesting active allocation of resources on the leaves [147]. In the leaves and roots of AGS2038, 30 and 28 metabolites, respectively, were changed, suggesting that the resource allocation in the sensitive genotype was suboptimal, favouring roots more than leaves [147]. Furthermore, in LA54, the accumulation of valine, tryptophan, malic acid, fumaric acid and citric acid showed a higher accumulation in the leaf compared to the root. These findings suggested that the differential allocation of resources in tolerant and susceptible genotypes was decisive in imparting tolerance to drought stress in wheat [147]. The finding that the differential allocation of amino acids, phenolics, alkaloids, flavonoids and organic acids is an important key mechanism in differentiating drought-resistant and susceptible varieties was further reinforced by ultra-performance liquid chromatography-mass spectrometry (UPLC-MS) [41].

Furthermore, to assess the effect of Cd toxicity on wheat metabolome, the metabolome profile following Cd application was investigated from two hexaploid wheat genotypes, AK58, and ZM10, with a low and high Cd-accumulation in grains, respectively [148]. Compared to ZM10, AK58 was found to have a greater root antioxidant system and higher levels of Cd bound to root cell walls, owing to the increased accumulation of hemicellulose and pectin to aid Cd binding. To further our understanding of Cd toxicity, ameliorating effect of Boron (B) was analysed on wheat growth following exposure to Cd treatment [149]. It was found that plant growth under Cd stress was adversely affected and B application was not able to fully recover the plant [149]. However, following the B application, accumulated Cd and malondialdehyde levels in the shoot and root decreased significantly. In addition, B application led to a reduction in the activity of enzymes, such as SOD and peroxidase (POD), that were induced in response to Cd stress [149]. Also, B application following exposure to Cd stress resulted in an increased accumulation of citric acid, galactaric acid, D-glucose and N6-galacturonyl-L-lysine and decreased accumulation of C16 sphinganine and threoninyl-tryptophan [149]. Hence, a significant role of galactose, sphingolipid, linoleic acid and propanoate metabolism and glycolysis/gluconeogenesis pathways was slated to alleviate Cd toxicity by inhibiting Cd uptake, changing metabolic profile and augmenting antioxidant activity.

## 5. Proteomics Approaches

Proteins, along with their post-translational modifications, are crucial for plant stress responses. Proteomics studies, therefore, provide valuable information about the cellular pathways involved in stress adaptation and mitigation. Initially, the term ‘proteomics’ referred to the methods used to analyze numerous proteins at a time; however, the term has now expanded to include any approach that provides information on the abundance, properties, interactions, activities or structures of proteins in a sample [150]. The analysis of the protein profile of wheat plants in response to abiotic stresses, such as drought, salinity and heat, is well documented (Table 3).

Molecular mechanisms in response to drought were investigated in two wheat lines, Zhongmai 8601 and YW642, drought tolerance and Thinopyrum intermedium 7XL/7DS translocation line, respectively, under a drought stress environment [158]. Two-dimensional difference gel electrophoresis (2D-DIGE) was employed to explore the differential accumulation protein (DAP) after 20 days of post anthesis (DPA) and a total of 146 DAPs were identified [158]. Furthermore, MALDI-TOF/TOF-MS was employed to identify the 113 unique proteins connoted by these DAPs [158]. Of these, 48 unique proteins exhibited upregulation and were involved in plant stress response, protein metabolism and, energy metabolism. Most significantly, 14 DAP genes were reported to have high expression levels in the 7XL/7DS translocation line during grain development periods [158]. Of these, four genes were identified as responding to drought stress, of which two genes showed oxidoreductase and dehydrogenase activities. In addition, three genes with potential protein binding, catalytic and transmembrane transporter-type roles were identified under drought and heat stress [158]. Functional relevance of a plant growth-promoting rhizobacterium (PGPR) Enterobacter cloacae SBP-8 was assessed under excessive salinity (200 mM NaCl) stress by investigating proteome profiles in the bacterial-inoculated wheat plants with and without salt stress [169]. A total of 286 differentially expressed proteins (DEPs) were identified and the majority of them were linked to metabolic pathways, photosynthesis and stress mechanisms [169]. Furthermore, bacterial inoculation was found to upregulate the expression of the Hsp70, Hsp90 organizing protein and cold shock protein CS66 at 200 mM NaCl stress [169].

To understand the effects of heat stress on wheat plants, proteome changes were analysed in the wheat kernel in the winter wheat cultivar Gaocheng 8901 under heat stress by iTRAQ (isobaric tags for relative and absolute quantitation) [161]. The iTRAQ analysis revealed quantitative information on 2493 proteins in the cultivar Gaocheng 8901 under heat stress, of which 116 were upregulated and 91 were downregulated [161]. A group of 78 DEPs were coupled to 83 KEGG signalling/metabolic pathways. Five DEPs, Elicitor responsive gene 3 (ERG3), brassinosteroid-insensitive 1 (BRI1), chaperone protein (CLPB1), histone cell cycle regulator (HIR1) and pre-mRNA processing factor (RSZ22), were involved in protein–protein interaction networks and were suggested to significantly impact the yield and quality of wheat grain under heat stress [161]. The relevance of stress-associated active proteins (SAAPs) involved in the process of the terminal heat tolerance of wheat has also been investigated [162]. The wheat *cvs.*, HD2985 (heat-tolerant) and HD2329 (heat-sensitive) were studied under heat stress at 38 °C for 2 h to identify 4271 SAAPs by employing iTRAQ. Under heat stress, 2800 and 2225 differentially expressed SAAPs were upregulated and downregulated in the tolerant cultivar HD2985, while 800 and 3600 expressed SAAPs were upregulated and downregulated in the sensitive cultivar HD2329, respectively [162]. Using the gene ontology analysis, differentially expressed SAAPs were characterized into three majorly functional groups, namely, molecular functions (51%), biological processes (39%), and cellular components (10%) [162]. SAAPs have been classified to regulate in defence- and stress-related activities. However, expression of ribonuclease TUDOR-1, HSP90, HSP20, peroxidase and HSC70 has been found in the wheat *cv*. HD2985 (heat-tolerant), and downregulation was observed in the wheat *cv*. HD2329 (heat-susceptible) under heat stress [162]. In addition, here was an increase of 8.2 folds *HSP17* and 2.2 folds of *CDPK* in expression under heat stress at 38 °C for 2 h in the spike of the wheat *cv.* HD2985 [162]. Likewise, there was a 22.5 folds *HSP17*, 4.5 folds *RuBisCo*, 4.3 folds *Rca* and 4.1 folds *OEEP* increase in expression in the leaves of the wheat cv. HD2985 under heat stress at 38 °C [162].

The Molecular mechanism of pathogenesis on the protein level in a wheat cultivar (Dongxuan 3) was analysed for two diseases, i.e., dwarf bunt (*Tilletia controversa* Kühn) and common bunt (*T. foetida* Kühn) [170]. The iTRAQ and Ultra-High-Performance Liquid Chromatography (UHPLC)-MS/MS analysis were employed for detecting DEPs [170]. A total of 4553 DEPs following infection by *T. controversa* and 804 DEPs following infection by *T. foetida* were identified [170]. Of the 4553 and 804 DEPs, 4100 and 447 were upregulated and were linked with metabolic process, catalytic activity, transferase activity, photosynthetic membrane and oxidoreductase activity [170].

Molecular mechanisms of the interaction between the *Fusarium pseudograminearum* WZ-8A and two wheat cultivars were explored [171]. Crown rot pathogen *F. pseudograminearum* was inoculated into UC1110, disease-susceptible and PI61750, disease-resistant wheat cultivars. Three days after inoculation (DAI), the average root diameter and malondialdehyde content of the roots was found to decrease, and the number of root tips increased [171]. To determine if the morphological, physiological and biochemical responses of the roots to disease differed between the two cultivars, Tandem Mass Tag (TMT) labelling and LC-MS/MS were employed [171]. Using TMT quantitative proteomics analysis, 366 DEPs were identified, of which 163 were from UC1110. Gene ontology and KEGG analysis identified candidate genes involved in biological processes, such as phenylpropanoid biosynthesis, photosynthesis and glutathione metabolism and cellular components, such as ribosomes [171].

## 6. Multiomics Approaches

Since the advancements in omics technologies and computational tools, the use of a multiomics approach has become a major area of thrust to answer burning questions in the stress biology of a crop and to reduce the number of false positives arising from the use of a single data [172,173]. It has been proposed that, by inspecting the change in correlation in the transcript–protein–metabolite between the control and stress conditions, biological processes strongly regulated by the plants can be recognised.

Various web-based tools and visualization portals are available to analyze the multi-omics datasets, such as PAINTOMICS, KaPPA-view, COVAIN, and O-miner [174,175,176,177,178]. In PAINTOMICS, the integrated visualization of transcriptomics and metabolomics datasets is possible and displays the data on KEGG pathway maps. The KaPPA-view tool allows the integration of transcript and metabolite data on plant metabolic pathway maps. The COVAIN tool offers statistical analysis of the integrated omics dataset through the KEGG pathway and gene ontology analysis [177].

A combined phenotypical, molecular and metabolomic approaches were employed to understand the role of symbiosis with arbuscular mycorrhiza (AM) on the mineral nutrition of wheat and in response to pathogen attack by the fungus *Xanthomonas translucens* [4]. The changes in transcripts and proteins in the roots and leaves of wheat plants with AM and with AM along with pathogen infection were investigated. It was observed that the transcriptomic and proteomic datasets shared 3.7 and 0.9% and 19 and 20% of differentially expressed genes (DEG) and differentially expressed proteins (DEP), respectively [4]. Several genes involved in nutrient uptake, primary metabolism and phytohormone regulation were found to be differentially regulated between the lines with and without AM [4]. Most significantly, the homologies searches identified important orthologous candidate genes, such as coding for Glycerol-3-phosphate acyltransferase (*OsRAM2* homolog), LysM domain-containing protein (*OsLysM* homolog) and *ABC-2 type transporter* (*OsSTR1* homolog) [4]. In addition, the transcription profiles of some phosphate transporters (PTs) were investigated by RNA-seq and qRT-PCR analyses, such as *TaAMT3.1*, *TaSulfTr2*, *TaAKT1*, *TaPT10*, *TaPT11* and *TaPT12*. All of these PT genes were strongly induced in wheat-AM plants’ roots compared to plants without AM [4]. A total of 29 novel genes were found to be exclusively expressed in the leaves of AM plants known to be involved in the responses to biotic stress, including a pathogenesis-related protein *PR-1*, mildew resistance locus (*MLO*) genes and putative homologs of *RPM1* (Resistance *to Pseudomonas syringae* pv. maculicola 1) protein [4].

Both proteomics and metabolomics approaches were employed for two spring-wheat cultivars, Bahar and Kavir, drought-tolerant and drought-susceptible, respectively, to understand the underlying biochemical networks at play in wheat leaves under drought stress [154]. Metabolomic analysis revealed that the levels of primary metabolites, such as amino acids, sugars and organic acids, were found to change in response to water deficiency [154]. In the Bahar cv, the accumulation of branched-chain amino acids, lysine, proline, aromatic, arginine and methionine was found in response to drought stress, in addition to the activation of shikimate pathway-mediated tryptophan accumulation aiding auxin production [154]. In the Kavir metabolome, only two pathways were found to be significantly affected in response to stress, one being of purine metabolism [154]. Only a few alterations in the metabolomic profile of Kavir were potent enough to induce susceptibility to drought stress, suggesting that unravelling the subtle changes in complex pathways leads to profound phenotypical changes [154].

Combined metabolomic and proteomic approaches was employed to dissect the resistance mechanism conferred by Fhb1 QTL in the near-isogenic lines (NILs) derived from the wheat genotype Nyubai [142]. The comparison of the metabolomic and proteomic profiles in the NILs revealed that the shunt phenylpropanoid pathway-producing metabolites, such as hydroxycinnamic acid amides, phenolic glucosides and flavonoids, played an important role [142]. Using confocal microscopy, it was confirmed that cell wall thickening, due to the deposition of hydroxycinnamic acid amides, phenolic glucosides and flavonoids, was responsible for imparting resistance rather than the conversion of DON to less toxic deoxynivalenol 3-O-glucoside, demonstrating alternate novel pathways that could play a pivotal role [142].

Large-scale multi-omics analysis was used to dissect the wheat stem solidness and resistance to wheat stem sawfly (WSS) [179]. A combined transcriptomic, metabolomic and proteomic approach was deployed on two wheat cultivars, the solid-stemmed Choteau and semi-solid-stemmed Scholar, differentiating for a QTL identified previously on chromosome 3B for solid stem [179]. The semi-solid-stemmed cultivar showed a differential regulation of 15 transcripts on WSS infection, of which 5 were upregulated and coded for an auxin efflux carrier component, pathogenesis-related (PR) genes 5 (CPR-5) protein, NADH dehydrogenase (NDH-A) and a magnesium transporter [179]. The solid-stemmed Choteau variety, on the other hand, showed nine DEGs, of which only one was found to be upregulated, while the remaining transcripts were downregulated [179]. In silico analysis revealed that this upregulated DEG coded for a cysteine-rich receptor-like protein kinase. The proteomic and metabolomic data further suggested the activation of the phenylpropanoid and pentose phosphate pathways in response to WSS infestation [179]. The key metabolites of the phenylpropanoid pathway are flavonoids and lignin. Lignin is is known to protect plants against mechanical damage under stress conditions, including drought or wounding, and acts as a physical barrier against pathogens [180].

Molecular mechanisms underlying the adaptation of wheat were determined in response to potassium (K) deficiency by investigating the transcriptomes and metabolomes of a panel of wheat accessions differing in K-deficiency tolerance [181]. The panel was subjected to a low-K treatment under hydroponic culture conditions for 14 days and root samples were collected for transcriptomic and metabolomic analyses [181]. It was found that the three CIPK (serine/threonine protein kinases)-encoding DEGs, i.e., CIPK14, CIPK9 and CIPK27, were upregulated in KN9204 and unchanged in BN207, while the expression of the three other DEGs (CIPK19, CIPK15 and CIPK29) was downregulated in BN207. These CIPKs were suggested to be underlying candidates for low-K tolerance. Recently, a multiomics approach was employed to identify the genes conferring a dense spike in a wheat-dense spike mutant (wds) obtained from a landrace Huangfangzhu [128]. Two large deletion segments on chromosome 6B, at 334.8–424.3 and 579.4–717.8 Mb, in the wds mutant and 499 genes were identified within the deleted regions [128]. The candidate gene TraesCS6B01G334600, for which an ortholog was identified in rice (*OsBUL1*), is known to regulate spike length and grain length [182]. This study, therefore, provided a basis for using this gene in wheat breeding for yield improvement.

In most of the GWAS studies, the biological mechanisms underlying the associations are unknown. To deal with this, GWAS has been integrated with metabolite profiling and/or gene regulatory network or pathway analysis and this approach has proven to be very effective in increasing the utility of information gained from GWAS [183,184,185]. A combination of GWAS and SNP-phenotype network approaches was employed in a core collection of wheat to identify the genetic basis of grain yield and spikelet-fertility-related traits [185]. The 118 SNPs that were found to be significant in their study at a false discovery rate (FDR) = 0.01 were used in the genotype–phenotype network analysis, of which 14 SNPs directly interacted with many of the agronomic traits. In silico analysis of these 14 SNPs revealed the strong involvement of metal ion transport and Gibberellin 2-oxidases (GA2oxs) genes in controlling the spikelet sterility [185]. Additionally, these genes showed a higher expression in the grain and spike, which further suggested their pivotal role in controlling the traits [185]. Acombination of GWAS and gene co-network analysis was used to dissect the genetic basis of root traits in a panel of winter wheat [80]. Three genomic regions on chromosomes 6A, 6B and 6D controlling the rooting depth, canopy temperature and yield were identified [80]. Using a minirhizotron system, accessions with favourable marker alleles of peak SNPs on chromosomes 6A, 6B and 6D were found to have longer roots than those with alternative marker alleles at these loci. Most significantly, 13 *NRT2* genes from a nitrate transporter gene family were identified in the study, which are known to improve nitrogen use efficiency and yield in rice [186].

The three rust pathogens, *Puccinia triticina* (leaf rust), *Puccinia graminis* f. sp. *tritici* (stem rust) and *Puccinia striiformis* f. sp. *tritici* (stripe rust), are the most damaging pathogens in wheat and create massive losses in yield [187,188,189]. The constant appearance of their new races in wheat-growing areas and their high adaptability to diverse climates pose serious challenges in developing resistance in wheat against rust. Although genomic selection (GS) has been deployed previously to improve predictions for fusarium head blight resistance, an integration of GS and transcriptomics data was suggested to improve the applicability of GS for rusts [190]. To be effective in a practical breeding program, the transcriptomes of only a limited number of founder lines could be integrated to develop GS models for imputing the value of others by using pedigree and genomic data [191].

## 7. Conclusions and Future Perspectives

Diverse omics tools are now at hand to understand the quantitative and qualitative reaction of wheat to various stresses and to pinpoint the potential candidate genes for marker-based breeding and/or for gene manipulation by CRISPR-based genome editing. The combined application of genomics and transcriptomics approaches has unveiled gene sets of certain gene families that could be deployed for generating stress-tolerant varieties. Furthermore, metabolomic and proteomic approaches have revealed additional candidate genes, gene networks and mechanisms, suggesting that a network of diverse gene actions comes into play under a stress condition. For example, genomics and transcriptomics studies demonstrated the role of ROS enzymes to be key players, while metabolomics studies highlighted the important roles played by amino acid transporters in the differential allocation of resources under drought stress.

While genomics and transcriptomics approaches are routinely employed in wheat for in-depth analysis, the combined application of metabolomics and proteomics has not been explored to a great extent. This is partly due to the affordability of genomics and transcriptomics data on large panels owing to low sequencing costs, as compared to metabolomics and proteomics, and due to the availability of user-friendly statistical and bioinformatics tools facilitating the analyses of genomics and transcriptomics. We envision that similar advancements in metabolomics and proteomics tools in the future will enable the comparative analysis of data from different streams of omics platforms in wheat. A straightforward application combining GWAS with proteomics (pGWAS) and metabolomics (mGWAS) would become feasible to identify the novel genes and functional pathways underlying complex traits. In addition to unveiling the novel candidate genes, the combined analysis of data will help in narrowing down the number of candidate genes to be subjected to functional validation by utilizing the mutant resource freely available to wheat scientists. The information of sequenced mutant libraries in the Ensemble database (http://plants.ensembl.org/Triticum_aestivum/Info/Index) and SNP-based primers available to screen mutants have opened vistas for conducting a plethora of novel functional genomics studies in wheat. Furthermore, amalgamating omics approaches shall pave the way for developing mathematical and GS models to predict plant performance under adverse conditions. This will facilitate wheat breeders to select lines with suitable gene-trait combinations to improve crop productivity under stress conditions. The synchronous analysis of genome, transcriptomes, proteome, metabolome and correlation of findings is, therefore, highly relevant and timely to develop efficient crop improvement programs in wheat.

## Figures and Tables

**Figure 1 plants-12-00426-f001:**
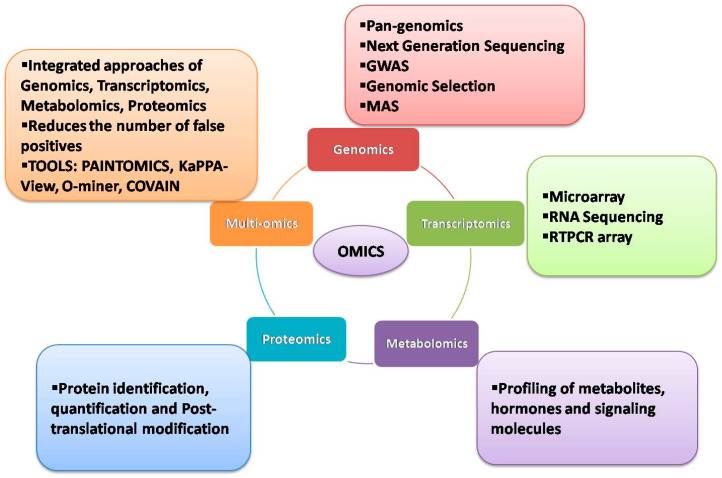
The schematic representation of various omics approaches deployed in crops.

**Table 1 plants-12-00426-t001:** Representative studies of gene families with putative roles under biotic and abiotic stresses in wheat.

Gene Family	Putative Annotation	Stress	Reference
**Biotic stress**			
LIM (Lin-11, IsI-1 and Mec-3)	Transcription factors	Fusarium head blight	[51]
bHLH (basic helix-loop-helix)	Transcription factors	Fusarium head blight and Septoria tritici blotch	[52]
Serpin (serine protease inhibitor)	Protease inhibitors	Fusarium head blight	[53]
SNARE	Transport proteins	Powdery mildew	[54]
SWEET	Sugar transporter	Stem rust	[55]
CNGCs	Calcium channel	Stripe rust	[56]
AP (Aspartic proteases)	Proteolysis enzymes	Powdery mildew	[57]
ZF_HD (Zinc finger homeodomain)	Transcription factors	Fusarium head blight	[58]
Xylanase inhibitor	Plant pathogen interaction	Fusarium head blight	[59]
Caffeoyl-coenzyme A O-methyltransferase	Lignin biosynthesis	Fusarium head blight	[13]
**Abiotic stress**			
DEAD box RNA helicases	RNA metabolism	Drought, cold and salt	[60]
WRKY	Transcription factors	Drought and salt	[61]
Domain of unknown function	Uncharacterized	Salt	[62]
Trihelix	Plant specific transcription factors	Salt and cold	[63]
HSPs (Heat shock proteins)	Protein folding	Heat	[64]
NAC (NAM-ATAF1-2-CUC2)	Plant specific transcription factors	Heat and drought	[65]
bZIP (basic leucine zipper)	Transcription factors	Heat, salinity, drought and oxidative stress	[66]
ASR (ABA-stress-ripening) genes	Transcription factors	Salt and low temperature	[67]
SRO (similar to radical-induced cell death 1 proteins)	Small protein family	Various stresses	[68]
PLC (Phospholipase C)	Cytoplasmic membranes	Salt, low temperature and drought	[54]
Expansin	Cell wall component	Salt	[69]
BAM (B-amylase)	Sugar	Heat and drought	[70]
LIM (Lin-11, IsI-1 and Mec-3)	Transcription factors	Heat, drought, salt, abscisic acid	[51]
Growth regulating factors	Transcription factors	Osmotic stress	[71]
Amino acid transporter	Transporter proteins	Heat and drought	[72]

**Table 2 plants-12-00426-t002:** Transcriptomic studies on analyzing stress responses in wheat.

Trait for Transcriptome Analysis	Summary	Reference
Heat and drought stress	HSFs and DREBs are involved in alleviating stress effect. Further, 1328 TFs found to be responsive to stress treatment.	[111]
Salt stress	TF’s including WD40-like, C2H2, MYB-HB-like, genes coding for V-ATPase, glutathione S-transferases, cytochrome c oxidase and Cbl-interacting protein kinasewere over expressed in *Triticum aestivum* cv. Kharchia Local.	[112]
Drought and/or heat stress	Drought-responsive WRKY transcription factor genes *TaWRKY1* and *TaWRKY33* were found to confer drought and/or heat resistance in Arabidopsis.	[113]
High temperature stress	Identified six heat-induced MYB genes in wheat.	[46]
Metal stress	The expressions of ABC transporters in dwarf polish wheat played important roles in metal transport (Cd, Cu, Mg, Zn, Fe, and Ni) and detoxification.	[114]
Drought stress	Drought stress significantly upregulated auxin receptor (AFB2) and ABA responsive transcription factors (MYB78, WRKY18 and GBF3), ACC oxidase and 2OG-Fe(II) oxygenase in roots. Genes related to gibberellic acid, jasmonic acid and phenylpropanoid pathways were down regulated in roots.	[115]
Water stress	Root transcriptome profiles identified DEGs involved in carbon metabolism, flavonoid biosynthesis and phytohormone signal transduction.	[6]
Leaf rust	Genes involved in reactive oxygen species (ROS) homeostasis and several genes encoding TFs, most abundant being WRKY TFs, were identified along with some ncRNAs and histone variants in HD 2329 + Lr28 NIL in comparison to HD 2329.	[116]
Seedling salt stress	Salt tolerance was conferred by polyunsaturated fatty acid (PUFAs) by enhancing the photosynthetic system and JA-related pathways.	[45]
Leaf spot (*Bipolaris sorokiniana*) tolerance	The upregulation of hydrolase inhibitor, NAC (including *NAM*, *ATAF1* and *CUC2*) transcriptional factor, and peroxidase in infected wheat tissues suggested their central roles in the defensive response of wheat to *Bipolaris sorokiniana*.	[117]
Elevated CO_2_ and high temperature stress	DEGs in response to stress includes protein kinases, receptor kinases, and transcription factors.	[118]
Stripe rust	Several regulators, including splicing and transcription factors and Hsp70 protein are responsive in *Puccinia striformis* induced response network.	[119]
Water Stress	Comparative analysis of root transcriptome revealed that transcription factors, pyroline-5-carboxylate reductase and late-embryogenesis-abundant proteins were upregulated genes in the tolerant cultivar.	[120]
Heat stress during grain filling	Hsp-family, ascorbate peroxidase, β-amylase, γ-gliadin-2 and LMW-glutenin were heat stress responsive and were upregulated during stress.	[121]

**Table 3 plants-12-00426-t003:** Proteomic studies on analyzing stress responses in wheat.

Abiotic Stresses Analysed	Research Summary	Reference
Drought Stress	Stress response of two genotypes SW89.5193/kAu2|SERI M 82 (Susceptible|Tolerant) was analyzed under drought. They found 40 roots; 73 leaves differentially expressed proteins (DEPs) in roots and leaves, respectively.	[151]
	Pretreatment with 0.5 mM salicylic acid (SA) for 3 days significantly enhanced the growth and tolerance to subsequent drought stress in Yumai 34. A total of 76 proteins were found to be differentially regulated by using 2-DE, MALDI-TOF-TOF from leaf samples.	[152]
	Monitored the roots of two different wheat varieties, Nesser (drought-tolerant) and Opata (drought-sensitive), in the absence and presence of abscisic acid (ABA, as a proxy for drought). A total of 151 proteins were found to be differentially regulated.	[153]
	Monitored the stress response of two cultivars Bahar, drought-susceptible; Kavir, drought tolerant under drought stress. A total of 81 proteins were found to be differentially regulated.	[154]
	Monitored the stress response of two varieties, Ningchun 4 (Tolerant) and Chinese Spring (Susceptible), at grain development stage: A total of 91 proteins were found to be differentially regulated.	[155]
	Monitored *PEPC* transgenic lines for drought tolerance, expressing maize C4 phosphoenolpyruvate carboxylase (PEPC) gene. By employing the 2-DE, MALDI-TOF, a total of 75 genes were found to be differentially regulated from flag leaf samples.	[156]
	Monitored the stress response of two wheat cultivars, Xihan No. 2 and Longchun 23, under dehydration and rehydration. They reported 84 and 64 proteins differentially regulated in Xihan No. 2 and Longchun 23, respectively.	[157]
	Monitored the stress response of two varieties, Zhongmai 8601 and Zhongmai 8601-Thinopyrum intermedium 7XL/7DS translocation line YW642, under drought stress at grain development stage. They found the differential regulation of 146 proteins in response to drought.	[158]
	Monitored the NaHS treated seedlings under drought stress in Yumai 34. They found the differential regulation of 120 proteins.	[159]
	Identified drought-tolerant proteins via virus-induced gene silencing in drought-tolerant XN979 and drought-sensitive LA379 varieties. They found the differential regulation of 335 proteins in response to stress.	[160]
Heat Stress	Monitored the stress response of Gaocheng 8901, winter wheat. They found the upregulation of 207 proteins.	[161]
	Monitored the stress response of two cultivars, HD2985 (thermotolerant) and HD2329 (thermosusceptible), at pollination and grain filling stages. They identified 4271 stress-associated proteins.	[162]
	Monitored the BWL4444 (HD2967+ Yr10) plants during the grain filling stage to understand the effect of heat stress in heat-tolerant varieties. Differential expression of 153 proteins was found in developing grain samples.	[163]
	Monitored the stability of the filling rate under heat stress in two wheat varieties, Chinese Spring and Liao-10. They found 309 proteins associated with heat stress.	[164]
Salinity Stress	Monitored priming-induced salt tolerance on vigor in *T durum* var. Waha. The ratio of seed weight to the volume of solution employed for priming was 1:5; 12 h soaking. Priming treatments: distilled water (c) and ascorbate (t). Salt stress: 10 mL of saline solution (NaCl 250 mmol L^−1^) or distilled water (control) and drying under shade with forced air at 27 ± 3 °C. They found 72 proteins hydroprimed and 83 proteins ascorbate primed.	[165]
	Monitored the roots of two wheat varieties, Jing-411 (salt-tolerant) and Chinese Spring (salt-sensitive), under salt stress. They found 52 proteins in Jing-411 and 47 proteins in Chinese Spring to be differentially regulated.	[166]
	Monitored the roots of two wheat varieties, Kharchia-65 (highly salt-tolerant) and PBW-373 (salt-sensitive), under salt stress. They found 2520 proteins in Kharchia-65 and 1633 proteins in PBW-373 to be differentially regulated.	[167]
Cold Stress	Monitored the cold stress response of seeds of one wheat cultivar, *T urartu* L. They found 34 proteins differentially regulated in response to cold stress in leaf samples.	[168]

## Data Availability

Not applicable.

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
