# Peer review of "Wheat Omics: Advancements and Opportunities"

_plants, 2023, doi:10.3390/plants12030426_

Round 1
Reviewer 1 Report
The authors well summarize the results obtained with various omics technologies, such as genomics, transcriptomics, metabolomics and proteomics. Therefore, I consider that the results summarized can be used in wheat to enhance understanding of the stress biology of the crop and the molecular mechanisms underlying stress tolerance. However, I have several major and minor comments about the results as described below. So, I recommend the authors to revise the manuscript according to my comments, if my comments are proper.
Major comments
1. It is described in the manuscript that Amirbakhtiar et al. (2019) identified a total of 26,171 novel transcripts, of which 5,128 were found to be differentially expressed in response to salinity stress, and that most significantly, the authors observed increased expression levels of TaABAC15, Ta.ACA7, Ta.ANN4, Ta.NCL2, and 48 MYBs in the salt-tolerant wheat cultivar in comparison to the salt-susceptible cultivar. Furthermore, it is also described that TraesCS2A02G395000 that codes for an oxoglutarate/iron-dependent dioxygenase was characrterized central in controlling salinity stress in wheat. Additionally, two more genes coding for copper amine oxidase and an amino acid transporter were identified. Nevertheless, for example, specific changes of both TaABAC15 and an oxoglutarate/iron-dependent dioxygenase are not observed in other studies carried out under similar salinity stress.
Although I read the manuscript, I could not find out examples showing that the same genes or proteins were similarly regulated under the same stress, such as drought stress, salinity stress, heat stress and so on. Therefore, I consider that the authors should explain the reasons why the same genes or proteins were not similarly regulated under the same stress and/or that the authors should show perspectives of unified interpretation about the same or similar research results, for example, before or in Conclusions and Future Perspectives, if possible.
Minor comments
1. 7th line from the top of Introduction: One word “transcriptomics” should be deleted, because the word was used twice in the same sentence.
2. 5th line from the top of Wheat genomics: A phrase “single nucleotide polymorphism” should be added in front of “(SNP)”, which is used for the first time in the manuscript. Accompanied by the addition, the phrase on 5th line from the bottom of Page 5 should be deleted.
3. Amirbhakhtiar et al., 2021 on 8th line from the bottom of Page 8, Guo et al., 2020 on third line from the top of Page 11, Qin et al., 2022 on 4th line from the bottom of Page 11 and Michaletti et al., 2018 on 5th line from the top of Page 16 should be revised as Amirbhakhtiar et al., (2021), Guo et al., (2020), Qin et al., (2022) and Michaletti et al., (2018), respectively.
4. A phrase “reactive oxygen species” should be added in front of ROS as reactive oxygen species (ROS) on 11th line from the bottom of Page 7. Accompanied by the addition, the phrase on 10th line of Page 9 should be modified as ROS.
5. Comma “,” after the phrase “in the tolerant variety” should be period “.”.
6. (Table ...) after the phrase “different stresses” on 7th line from the top of Proteomics on Page 12 should be (Table 3).
Author Response
Response to Reviewer 1 Comments
Major Comments
Point 1: It is described in the manuscript that Amirbakhtiar et al. (2019) identified a total of 26,171 novel transcripts, of which 5,128 were found to be differentially expressed in response to salinity stress, and that most significantly, the authors observed increased expression levels of TaABAC15, Ta.ACA7, Ta.ANN4, Ta.NCL2, and 48 MYBs in the salt-tolerant wheat cultivar in comparison to the salt-susceptible cultivar. Furthermore, it is also described that TraesCS2A02G395000 that codes for an oxoglutarate/iron-dependent dioxygenase was characrterized central in controlling salinity stress in wheat. Additionally, two more genes coding for copper amine oxidase and an amino acid transporter were identified. Nevertheless, for example, specific changes of both TaABAC15 and an oxoglutarate/iron-dependent dioxygenase are not observed in other studies carried out under similar salinity stress.
Although I read the manuscript, I could not find out examples showing that the same genes or proteins were similarly regulated under the same stress, such as drought stress, salinity stress, heat stress and so on. Therefore, I consider that the authors should explain the reasons why the same genes or proteins were not similarly regulated under the same stress and/or that the authors should show perspectives of unified interpretation about the same or similar research results, for example, before or in Conclusions and Future Perspectives, if possible.
Response 1: We have made suitable corrections
Minor comment.
Point 1: 7th line from the top of Introduction: One-word “transcriptomics” should be deleted, because the word was used twice in the same sentence.
Response 1: The suggested change has been incorporated.
Point 2: 5th line from the top of Wheat genomics: A phrase “single nucleotide polymorphism” should be added in front of “(SNP)”, which is used for the first time in the manuscript. Accompanied by the addition, the phrase on 5th line from the bottom of Page 5 should be deleted.
Response 2: The suggested change has been incorporated.
Point 3: Amirbhakhtiar et al., 2021 on 8th line from the bottom of Page 8, Guo et al., 2020 on third line from the top of Page 11, Qin et al., 2022 on 4th line from the bottom of Page 11 and Michaletti et al., 2018 on 5th line from the top of Page 16 should be revised as Amirbhakhtiar et al., (2021), Guo et al., (2020), Qin et al., (2022) and Michaletti et al., (2018), respectively.
Response 3: The suggested change has been incorporated.
Point 4: A phrase “” should be added in front of ROS as (ROS) on 11th line from the bottom of Page 7. Accompanied by the addition, the phrase on 10th line of Page 9 should be modified as ROS.
Response 4: The suggested change has been incorporated.
Point 5: Comma “,” after the phrase “” should be period “.”.
Response 5: The suggested change has been incorporated.
Point 6: (Table ...) after the phrase “different stresses” on 7th line from the top of Proteomics on Page 12 should be (Table 3).
Response 6: The suggested change has been incorporated.

Reviewer 2 Report
Wheat Omics: Advancements and Opportunites
Sehgal et al, submitted to Plants 2022
The manuscript is a review on the status of different omics approaches used from wheat research, notably encompassing genomics, transcriptomics, metabolomics and proteomics, but also relating some studies that managed to combine two or more of these fields.
The manuscript is relatively well-written with a clear overall organization. The article could be of broad interest, for scientists involved in wheat research but not only. However, it suffers from several issues that need to be addressed:
- As a review, the manuscript is citing many studies, and on the section “Genomics” alone, at least 30 cited articles could not be found in the reference list. This list is also generally badly organized, with alphabetical order not systematically used. The missing references is also true for the other sections, while extra elements are present in the list even if not cited in the text. Please review and clean up entirely.
- The genomics section is relatively clear and makes several points on the improvement in the technologies leading to better understanding of the genome organization and content. However, from the transcriptomics to the proteomics section, the authors change their angle to a thorough but painful description of different studies. They describe at length many results without commenting on the impact of the omics technologies themselves on these results. The reader is left with a lot of information on different genes, metabolites or proteins involved in the response to different stresses. The multi-omics section is also in the same frame of mind, but with the added value of underlying the impact of combining approaches.
I would suggest to re-write all these sections to help scientists in identifying the different tools at hand and their best use.
- Finally, the conclusion is a little unsettling, the initial sentence pointing to a particular stage in wheat development out of nowhere, before stating that tools are available but are not used simultaneously, which should be the focus for the future, including bioinformatics tools to integrate the results faster. I would also suggest to re-write this paragraph to make it clearer.
Author Response
Response to Reviewer 2 Comments
Major Comments
The manuscript is a review on the status of different omics approaches used from wheat research, notably encompassing genomics, transcriptomics, metabolomics and proteomics, but also relating some studies that managed to combine two or more of these fields.
The manuscript is relatively well-written with a clear overall organization. The article could be of broad interest, for scientists involved in wheat research but not only. However, it suffers from several issues that need to be addressed:
Point 1: As a review, the manuscript is citing many studies, and on the section “Genomics” alone, at least 30 cited articles could not be found in the reference list. This list is also generally badly organized, with alphabetical order not systematically used. The missing references are also true for the other sections, while extra elements are present in the list even if not cited in the text. Please review and clean up entirely.
Response 1: The reference list is updated.
Point 2: The genomics section is relatively clear and makes several points on the improvement in the technologies leading to a better understanding of the genome organization and content. However, from the transcriptomics to the proteomics section, the authors change their angle to a thorough but painful description of different studies. They describe many results at length without commenting on the impact of the omics technologies on these results. The reader is left with a lot of information on different genes, metabolites or proteins involved in the response to different stresses. The multi-omics section is also in the same frame of mind, but with the added value of underlying the impact of combining approaches.
I would suggest re-writing all these sections to help scientists identify the different tools at hand and their best use.
Response 2: We have incorporated the suggestion to make the information provided more relevant with respect to the use of omics technologies at hand to dissect the regulatory player. We have done this for the sections mentioned.
Point 3: Finally, the conclusion is a little unsettling, the initial sentence points to a particular stage in wheat development out of nowhere, before stating that tools are available but are not used simultaneously, which should be the focus for the future, including bioinformatics tools to integrate the results faster. I would also suggest to re-write this paragraph to make it clearer.
Response 3: We have rewritten the paragraph mentioned.

Round 2
Reviewer 1 Report
The authors have extensively and appropriately revised the previous manuscript according to my major and minor comments. So, I would like to recommend the editor to accept the revised manuscript.
Author Response
We are thankful to the reviewer for his/her time and suggestions in making the manuscript reach its final shape.
Reviewer 2 Report
Wheat Omics: Advancements and Opportunites
Sehgal et al, re-submitted to Plants 2022
The manuscript is a review on the status of different omics approaches used from wheat research, notably encompassing genomics, transcriptomics, metabolomics and proteomics, but also relating some studies that managed to combine two or more of these fields. It has been re-submitted following a first round of peer-review in the same journal.
In my first review, I had three major points that needed addressing. The authors have made considerable efforts to do so and I wish to acknowledge this.
The reference list is much better now, which was critical for such a broad review.
The sections on transcriptomics, metabolomics and proteomics have been deeply re-written and are much easier to read now.
The conclusion has also been re-written in what I think is a good direction but could still be improved.
For example, the authors rightly state that the genomics and transcriptomics approaches are routinely used. I think one important point is the affordability of these techniques compared to metabolomics and proteomics. These have also become cheaper recently, but not as much as genomics and transcriptomics. So they cannot be deployed as widely yet.
The authors also cite a mutation mapping approach, but this is not referenced, so I am not sure what is meant. Mutations in hexaploidy wheat can be tricky with the possible need of combining mutations on all three genomes before seeing an effect on phenotype. CRISPR-based approaches can also be conducted for functional validation.
In the final part of the conclusion, the authors seem to be repeating the same statement in different sentences. “Tothiseffect,weenvisionthefocusoverthenextdecadeofwheatcropimprovementtocentrearoundthecomparativeanalysisofdatacomingfromdifferentstreamsofomicstudiessuchastranscriptomics,metabolomics,genomics,proteomicsandphenomics.Theareaofresearchthathasnotbeenexploredmuchinwheatandwillrequirespecificattentioninfutureiscombiningtranscriptome,metabolomeandproteomeprofilingwithgeneticmappingtonarrowdownthenumberofcandidategenesandpointoutthekeygenesinvolvedinunderlyingmechanismsofinterest »
I think an important point would be that for all this data to be explored and combined, it needs to fit the FAIR status defined by most data repository now. Metadata will probably be the key to avoid any mismatches between different datasets.
There are also minor corrections to be done. Unfortunately, the document I downloaded does not show any line numbers, so I can only refer to page numbers:
Page 8, “With the advancements in NGS, whole genome transcriptome analyses…”: delete genome to keep only whole transcriptome analyses
Page 11, “Furthermore, Luo et al (2021) explored the underlying regulatory network at play to improve tolerance to wheat with exogenous application of trehalose”: I am sure this is not tolerance to wheat but to a heat stress
Page 12, “A total of 15403 were found”: the word “genes” is probably missing
Page 12, “differential expression of 749 DEGs”: no need to repeat DE, DEGs should read “genes”.
Page 13, “TaHKT1 and TaKT was”: was should read were
Page 23, “Annunziata et al (2017) used a metabolomics approach and investigated the metabolome of durum wheat to investigate salinity tolerance”: switch to “… a metabolomics approach to investigate salinity tolerance in durum wheat”.
Page 23, “the uptake of sodium and its sequestration of sodium in the vacuoles”: change to “the uptake of sodium and its sequestration in the vacuoles”.
Page 23: “accumulation of valine…showed higher accumulation in the leaf”: change to “accumulation of valine… was higher in the leaf”
Page 27, “responding to drought stress, namely, of which two genes”: delete namely
Page 29, “Dongxuan E)”: delete the bracket.
Page 31, “by fungus” should be by the fungus
Author Response
Response 1: Response: We are thankful to the reviewer for further suggestions and have revised strictly according to the suggestions. Regarding the mutations part, we want to say that since this mutant resource is available in wheat and will KASP sequences, it will be much easier to validate the candidate genes that will be pointed out by the application of omics approaches. We have kept this in the Conclusions section but have slightly modified it to show the right perspective.